# Intelligent Random Access for Massive-Machine Type Communications in Sliced Mobile Networks

**Bei Yang [1], Fengsheng Wei [2,3,*] , Xiaoming She [1], Zheng Jiang [1], Jianchi Zhu [1], Peng Chen [1] and Jianxiu Wang [1]**

[1] China Telecom Research Institute, Beijing 102209, China
[2] Yangtze Delta Region Institute (Huzhou), University of Electronic Science and Technology of China, Huzhou 313001, China
[3] National Key Laboratory of Science and Technology on Communications, University of Electronic Science and Technology of China, Chengdu 611731, China
**\*** Correspondence: weifengsheng@csj.uestc.edu.cn

**Abstract:** With the emerging Internet of Things paradigm, massive Machine-Type Communication (mMTC) has been identified as one of the prominent services that enables a broad range of applications with various Quality of Service (QoS) requirements for 5G-and-beyond networks. However, it is very difficult to employ a monolithic physical network to support various mMTC applications with differentiated QoS requirements. Moreover, in ultra-dense mobile networks, the scarcity of the preamble and Physical Downlink Control CHannel (PDCCH) resources may easily lead to resource collisions when a large number of devices access the network simultaneously. To tackle these issues, in this paper, we propose a network slicing-enabled intelligent random access framework for mMTC. First, by tailoring a gigantic physical network into multiple lightweight network slices, fine-grained QoS provisioning can be accomplished, and the collision domain of Random Access (RA) can be effectively reduced. In addition, we propose a novel concept of sliced preambles (sPreambles), based on which the transitional RA procedure is optimized, and the issue of preamble shortage is effectively relieved. Furthermore, with the aim of alleviating PDCCH resource shortage and improving transmission efficiency, we propose a learning-based resource-sharing scheme that can intelligently multiplex the PDCCH resources in the naturally dynamic environment. Simulation results show that the proposed framework can efficiently allocate resources to individual mMTC devices while guaranteeing their QoS requirements in random access processes.

**Keywords:** massive-machine type communications; network slicing; random access; reinforcement learning

## 1. Introduction

Massive-Machine Type Communications (mMTC) is deemed as one of the fundamental services in the future digital world, which enables a broad range of applications in current 5G and the forth-coming Beyond 5G (B5G) networks, including industrial automation, intelligent transportation, smart-grid, etc. [1]. It has been acknowledged that the density of mMTC Devices (MTCDs) will be up to $10^6$ per km$^2$ in urban environments, which is significantly larger than that of Human Type Communication (HTC) devices [2]. According to the predictions made by Ericsson, the number of devices connected to communication networks will reach 26.9 billion by 2027, out of which more than 50% will be MTCDs [3]. As numerous MTCDs with diversified Quality of Service (QoS) requirements access the network, traditional HTC-oriented cellular networks face tremendous challenges.

The first key challenge originates from the heterogeneous QoS requirements of different mMTC applications. In a multi-application coexisting mobile network, the QoS requirements of different applications are distinct in terms of delay, reliability, mobility, energy consumption, etc. For instance, the transmission delay of the industrial alarming messages must be guaranteed, while the delay of periodic reporting in smart metering is

not critical. In addition, the underlying protocols that support these applications could be very different, such as the activation period, frame size, preamble format, etc. [4]. However, under traditional network architecture, the physical equipment is highly coupled with the services in the network, and it is thus very difficult to provide differentiated services for different mMTC applications. Consequently, a one-size-fits-all network cannot efficiently support differentiated QoS in multi-application coexisting mMTC scenarios.

Another major challenge for provisioning an mMTC service is incurred by the massive access requests from MTCDs. Existing contention-based Random Access (RA) protocols are mainly designed for HTC, and their performance deteriorates significantly in mMTC due to the limited RA resources. Among the resources involved in the RA procedure, there are two problems that restrict the performance of the contention-based RA protocols for mMTC devices. The first issue is the shortage of preamble resources, which are used to identify the MTCDs that initiate the RA request. Due to the limited available preambles in the network (only 54 preambles in a cell [5]), it is very likely that multiple MTCDs will initiate RA requests simultaneously, resulting in a high probability of preamble collision and an increase in the access delay. Another problem stems from the energy-efficient design objective of mMTC. In 5G-Advanced, limited resources are reserved for the Physical Downlink Control CHannel (PDCCH) to improve the transmission efficiency. It has been recognized that the limited PDCCH resources may severely constrain the access capability of the gNB [6].

Recently, there has been some pioneer work aimed at addressing both challenges. On the one hand, some researchers proposed to apply MTCD-grouping and priority parameters to provide a differentiated service [7–9]. However, these solutions cannot provide fine-grained service differentiation, and the operations involved rely heavily on the experience of the network operator. On the other hand, some researchers have proposed grant-free access mechanisms to simplify the traditional four-step handshake RA procedure [10–13]. In these solutions, the base station needs to predict the access time of all MTCDs in the cell, which is a computation-intensive task. Moreover, they perform poorly in the event-triggered mMTC applications due to the unpredictable nature of irregular stochastic events [14].

To address the aforementioned challenges, in this paper, we propose a network slicing-enabled RA framework for supporting mMTC services in sliced mobile networks. Network slicing enables multiple network slices (virtual networks) to share a single network infrastructure by exploiting the programmability, flexibility, and modularity of network software. Each network slice has specific functional characteristics that are customized by the service providers (i.e., the tenants) to meet specific QoS requirements. Thus, with the aid of network slicing, fine-grained service differentiation for various mMTC applications can be fulfilled. Moreover, by introducing network slicing, a monolithic physical network associated with massive MTCDs can be partitioned into multiple isolated lightweight network slices, thereby reducing the range of the *collision domain*. Consequently, the collision probability within each network slice can be drastically decreased, and thus the incurred access delay can be significantly reduced.

However, although the collision domain within each network slice is reduced, the available preamble and PDCCH resources are also reduced since the resources of a network slice are partitioned from the physical network. As a result, the reductions in the collision probability and the access delay may not be significant if the dynamic nature of the required mMTC service cannot be well addressed. In this paper, we propose a network slicing-enabled framework that incorporates a novel concept of *sliced preambles* (sPreambles) and an Actor-Critic Resource Sharing Scheme (ACRS). On the one hand, the introduction of sPreamble significantly expands the available preamble resources in each network slice, by which the collision probability as well as the access delay can be effectively decreased. On the other hand, our proposed ACRS scheme can achieve intelligent multiplexing of PDCCH resources between network slices, thereby improving their access capability efficiently. The main contributions of this paper are summarized as follows.

- We propose a network slicing-enabled mMTC random access framework, which leverages the customization capabilities of network slicing for the provisioning of differentiated QoS in multi-application coexisting mMTC scenarios.
- A novel concept of sPreamble is presented, which can scale the number of available preambles by a factor of $N$, where $N$ is the number of network slices deployed in the system.
- A reinforcement learning-based dynamic sharing scheme ACRS is proposed to intelligently allocate the PDCCH resources to individual network slices in dynamic environments. By using ACRS, the limited PDCCH resources can be effectively multiplexed by the network slices, thereby improving the access capability of the Radio Access Network (RAN).
- We verify the efficacy of the proposed framework through extensive numerical simulations. The simulation results demonstrate that the proposed framework can increase the access capability of the RAN and reduce the access delay significantly.

The remainder of this paper is organized as follows. In Section 2, we review the related work on mMTC access in sliced networks. In Section 3, we elaborate on the architectural design of the proposed framework. In Section 4, we present the system model and formulate the Dynamic PDCCH Resource Allocation Problem (DPRAP). In Section 5, we present the ACRS to solve the DPRAP in sliced mobile networks. In Section 6, we conduct numerical experiments to verify the effectiveness of our proposed framework. Finally, we conclude the paper in Section 7.

## 2. Related Work

In this section, we review the most relevant work on the problem of RA in mMTC. According to the mechanisms applied in RA, the existing research can be classified into three categories: coordinated RA, fast uplink grant (FUG)-based RA, and network slicing-enabled RA.

### 2.1. Coordinated RA

To alleviate preamble collisions caused by massive MTCDs simultaneously accessing the cellular network, 3GPP has adopted the Access Class Barring (ACB) scheme in RA. However, the access delay of the legacy ACB scheme increases sharply with the number of MTCDs, which restricts its application in ultra-dense mMTC networks. Thus, to tackle this issue, some improvements have been proposed for the original ACB scheme. Most of the improvements made on ACB are based on priority parameters and adaptive adjustment of ACB factors. In [9], the authors propose to classify the MTCDs into delay-sensitive and delay-tolerant groups. The delay-sensitive MTCDs are given the priority to access the network, while the ACB parameters of the delay-tolerant MTCDs are adaptively adjusted. Similarly, in [15], a priority-queuing-based ACB model is developed, and the modified ACB mechanism design is formulated as a noncooperative game with a unique Nash equilibrium. Recently, some researchers have proposes to exploit RL to adaptively tune the ACB parameters. Typically, based on Dueling Deep Q-Network (DDQN), the authors of [16] proposed a *dynamic ACB* solution to alleviate the delay issue of the legacy ACB scheme. In [5], a delay-aware priority access classification-based ACB mechanism is proposed. According to the delay requirements of the MTCD, an RL-based priority adjustment scheme is proposed.

### 2.2. FUG-Based RA

Although the preamble collision issue of the ACB scheme has been alleviated, the signaling overhead in such coordinated solutions is significant. The data size of the control signaling may be much larger than the size of user data in mMTC applications. To alleviate the signaling overhead, a number of FUG-based RA mechanisms are proposed. In FUG, only one signaling that represents an uplink grant to the MTCD is transmitted by the gNB [17]. To apply FUG in mMTC, the most prominent challenge is the optimal selection of

MTCDs to which the uplink grant is sent. Fortunately, recently emerging Machine Learning (ML) techniques, such as reinforcement learning, can automate the mMTC RA process by learning from historical data sets or interacting with the environments [4]. Thus, to improve the FUG mechanisms, there are a number of researchers that resort to ML techniques for the prediction of the RA requirements. Considering that the source traffic prediction algorithm cannot be perfect, the authors of [12] propose a sleeping Multi-Armed Bandit (MAB) framework for the scheduling of MTCDs using FUG to maximize a specific QoS metric, which is defined as a combination of the value of data packets, maximum tolerable access delay, and data rate. In [18], the authors propose a FUG-based RA mechanism based on Support Vector Machine (SVM) and Long Short-Term Memory (LSTM), where the MTCDs are first classified by SVM, and then their traffic is predicted by LSTM. In [13], the authors devise a two-stage RA technique that jointly exploits FUG and Non-Orthogonal Multiple Access (NOMA). To overcome the challenges of active MTCD prediction and scheduling, multi-armed bandit learning is adopted for the scheduling of fast grant MTCDs. Unfortunately, the aforementioned work does not exploit the benefits of network slicing; thus, they cannot provide fine-grained service differentiation for various MTCDs.

*2.3. Network Slicing-Enabled RA*

As one of the key enablers of 5G systems, network slicing can provide users with customized services by properly tailoring the physical network [19]. Compared with the priority-based RA mechanisms, network slicing can enhance the RA process by providing fine-grained service differentiation for heterogeneous MTCDs. Consequently, some recent work leverages the customization capability of network slicing to improve the RA performance for MTCDs. In [20], the authors propose a stochastic model of the sliced RAN for the HTC and mMTC coexistence scenario. Similarly, the authors of [21] present a detailed stochastic model of RA in sliced RAN and propose a Markov chain-based performance measurement technique for the HTC and mMTC coexistence scenarios. In [22], the authors investigate the integration of network slicing and mMTC application scenarios on campus.

However, these solutions do not jointly exploit the benefits brought by network slicing and ML. Thus, they are unable to provide fine-grained service differentiation for different MTCDs, resulting in unsatisfactory QoS provisioning for various applications in mMTC. To bridge this gap, we exploit both the customization capability of network slicing and the prediction capability of ML to guarantee heterogeneous QoS requirements of different MTCDs.

## 3. Network Slicing-Enabled RA Framework

*3.1. Architecture of Network Slicing-Enabled mMTC System*

The architecture of the proposed network slicing enabled random access framework for mMTC is shown in Figure 1. The architecture is composed of three layers: the infrastructure layer, the network slice layer, and the management-and-orchestration (MANO) layer. It should be emphasized that our framework is dedicated to the RA for mMTC service, which is different from the existing RAN slicing frameworks such as [23,24]. In particular, our framework mainly considers the resource slicing of the control plane rather than the data plane. The details of each layer are described in the following.

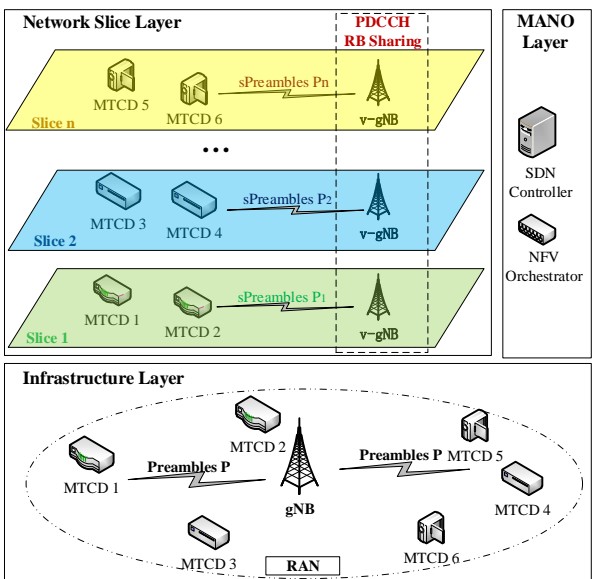

**Figure 1.** Architecture of the network slicing-enabled RA framework for mMTC.

### 3.1.1. Infrastructure Layer

Similar to traditional mobile communication networks, the infrastructure layer provides physical devices. In the underlying scenario, the infrastructure layer corresponds to a RAN that is composed of a gNB and multiple MTCDs with different QoS requirements. Within the RAN, several kinds of control plane resources are reserved to support the access of the MTCDs, among which the bottlenecks are the preamble resources and PDCCH resources. In 5G-Advanced, 54 preambles are available for contention-based access [25]. This number is much smaller than the number of MTCDs that access the network. Thus, preamble collision is inevitable in 5G-Advanced mMTC networks. In addition, as the only channel for the transmission of downlink control signaling in the RA procedure, the PDCCH monitored by MTCDs is severely resource-limited [6], which makes the gNB unable to respond to a large number of RA requests.

### 3.1.2. Network Slice Layer

At this layer, a number of network slices are created to provide differentiated services for different MTCDs. According to the QoS requirements of MTCDs, the physical resources of the physical infrastructure layer will be virtualized through the SDN/NFV technologies to form multiple network slices with typical QoS characteristics. Each network slice operates independently to serve a set of MTCDs with similar QoS requirements by using the virtual resources sliced from the infrastructure. Since this paper focuses on the RA procedure, this framework only deals with the slicing of the control plane. Specifically, the preambles of the infrastructure layer will be virtualized so that the number of available preambles in each network slice is the same as that of preambles in the infrastructure layer. In addition, the gNB will be sliced into multiple virtual gNBs (v-gNBs), each of which serves the MTCDs in the slice. Meanwhile, by dynamically sharing PDCCH resources between different slices, the bottleneck of PDCCH resources at each v-gNB can be efficiently alleviated.

### 3.1.3. MANO Layer

The MANO layer is responsible for the allocation and scheduling of the resources between network slices. This layer mainly consists of the essential elements for slice management and orchestration, including SDN controllers and NFV orchestrators. The SDN controller can dynamically allocate network resources and provide unified management and control, such as protocol operation, policy distribution, and link information collection.

The NFV orchestrators can allocate the virtualized physical resources to the network slices according to their instantaneous traffic load. Therefore, the MANO layer is the entity where the proposed ACRS scheme is executed. By leveraging the functionalities of SDN and NFV, each network slice can agilely share the PDCCH resource by interacting with the ever-changing traffic load in a trial-and-error manner.

### 3.2. Network Slicing-Enabled Random Access Procedure

In a sliced RAN, the RA procedure is similar to that in 5G-Advanced networks except that sPreamble and the ACRS schemes are applied.

### 3.2.1. The Concept of Sliced Preambles

Inspired by the idea of [26], we define the concept of *sliced preamble* to multiplex the preamble in the networks. In the proposed access scheme, an sPreamble is defined as the combination of the legacy preambles with *network slice instance identifier* (NsiId) [27]. With extra information incorporated, the number of available sPreambles is greatly increased. In particular, the number of sPreambles linearly scales with the number of the network slices. From the perspective of the infrastructure layer, the gNB can exactly differentiate the RA requests of the MTCDs from different slices according to their NsiId, although the same preambles are utilized. Thus, our proposed sPreambles can be applied in the sliced RAN with minor changes to the 5G-Advanced four-step handshake RA protocol.

### 3.2.2. The RA Procedure in the Sliced mMTC Network

In 5G-Advanced, two types of RA procedures are supported: 4-step RA type with MSG1 and 2-step RA type with MSGA, both of which support contention-based RA [25]. For the sake of simplicity and tractability, we only consider the 4-step RA and leave the hybrid 2-step and 4-step RA procedure for our future research. In our proposed framework the procedure of mMTC RA within each slice is illustrated in Figure 2. Before the RA procedure, the v-gNB will broadcast the available sPreamble to its associated MTCDs by SIB2 message, followed by the 4-step handshake procedure of the RA.

1. **Message 1:** In the first stage, each MTCD will send an RA request (Message 1), which contains a randomly selected sPreamble through the Physical Random Access CHannel (PRACH).
2. **Message 2:** If there are no collisions in the first step, the v-gNB replies to the MTCD with an RA response (RAR), which is Message 2 that includes an uplink grant and the Physical Uplink Shared CHannel (PUSCH) allocation information for the third step. The RAR is sent over the Physical Downlink Shared CHannel (PDSCH), which needs to be scheduled on the PDCCH [25].
3. **Message 3:** After successfully receiving the RAR from the v-gNB, the MTCD will send a connection request (Message 3) using the resource blocks announced by Message 2.
4. **Message 4:** The v-gNB sends a contention resolution (Message 4) to the MTCD through the PDSCH to indicate the success of the RA procedure. Again, the PDSCH needs to be scheduled on the PDCCH.

Since the channel resource for the transmission of Message 2 and Message 4 are scheduled by PDCCH, the available resources on PDCCH significantly affect the performance of RA. In our proposed framework, the PDCCH resources in each slice will be dynamically allocated through the ACRS scheme, which will be presented in the following sections.

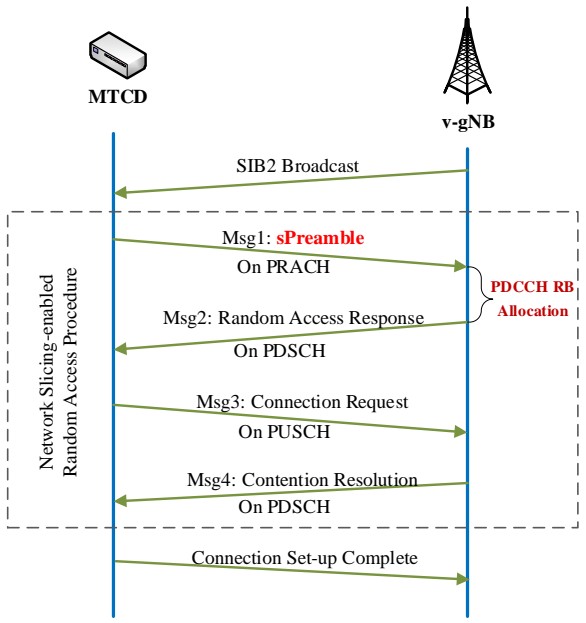

**Figure 2.** Network slicing-enabled random access procedure.

## 4. Dynamic PDCCH Resource Allocation Problem

*4.1. Network Model*

In this section, we formulate the PDCCH resource allocation problem under the aforementioned network slicing-based mMTC access scenario. In this paper, we focus on contention-based RA since the RA resources of a gNB, including the preamble and the PDCCH resources, are limited. For ease of reference, the key notations used in the paper are summarized in Table 1.

**Table 1.** Summary of notations.

| Notation | Description |
|---|---|
| **Parameters** | |
| $T$ | The duration of a time slot |
| $\mathcal{N}$ | Set of network slices |
| $\mathcal{M}^k$ | Set of MTCDs at time slot $k$ |
| $M^k$ | The cardinality of the set $\mathcal{M}^k$ |
| $\mathcal{M}_n^k$ | Set of MTCDs associated with slice $n$ at slot $k$ |
| $M_n^k$ | The cardinality of the set $\mathcal{M}_n^k$ |
| $n_j^k$ | The $j$-th MTCDs associated with slice $n$ |
| $p_n$ | The number of available sPreambles in slice $n$ |
| $R$ | The number of available CCEs in the RAN |
| $\tau_n$ | The time separation between two consecutive RAOs in slice $n$ |
| $\xi_n^k$ | The price paid by slice $n$ at the $k$-th time slot |
| $\rho_n^k$ | The priority factor of slice $n$ at the $k$-th time slot |
| **Decision Variables** | |
| $r_n^k$ | Integer variable indicates the number of CCEs allocated to slice $n$ at $k$-th time slot |

4.1.1. Physical Network Model

In our considered scenario, the time is slotted to $\{0, T, \cdots, kT, \cdots\}$. The $k$-th time slot corresponds to the time interval of $[(k-1)T, kT]$. We assume that there are $M$ MTCDs and one gNB in the infrastructure layer. Each MTCD has to access the gNB sporadically in order to transmit their data. The MTCDs are heterogeneous in terms of their battery capacity,

traffic characteristics and the QoS requirements, etc. Thus, a set of $\mathcal{N} = \{1, \cdots, N\}$ network slices with different Service Level Agreements (SLAs) are created to provide differentiated services for the MTCDs. At the $k$-th time slot, the number of MTCDs served by slice $n \in \mathcal{N}$ is denoted by $M_n^k$. Accordingly, the set of MTCDs in slice $n$ can be represented by $\mathcal{M}_n^k = \{n_1^k, \cdots, n_{M_n^k}^k\}$. It should be emphasized that the number of MTCDs served by each slice in each slot is time-varying, for the following two reasons.

- The MTCDs that fail to access the v-gNB before the current time slot will be added to the current MTCD set;
- Stochastic arrival and departure of MTCDs in each slice.

Thus, $M_n^k$ is varying between different time slots. Without loss of generality, we assume that each MTCD can be only served by one network slice. Thus, we have $\sum_{n \in \mathcal{N}} M_n^k = M^k$.

At the gNB side, the number of available preambles is denoted by $P$. In 5G, the PDCCH resources are measured in Control Channel Elements (CCEs) [6]. We assume that the number of available CCEs of PDCCH resources is $R$. In the RA procedure, the time at which the MTCDs are allowed to access the gNB is called RA Opportunities (RAOs) [28], which are broadcast by the gNB. The gNB broadcasts the periodicity of the RAO by the variable called the *Physical RACH Configuration Index*, which has 64 possible configurations [25]. In our framework, the RAO configuration within each slice is assumed to be different.

### 4.1.2. Network Slice

The gNB will be "sliced" to $N$ v-gNBs, each of which is responsible for supporting the RA of the MTCDs belonging to the slice. According to our proposed framework, the $P$ preambles will be abstracted to at most $NP$ sPreambles by the network slices. Without loss of generality, we assume there are $p_n$ sPreambles reserved for contention-based RA in slice $n$. For slice $n$, we assume that the time interval between two RAOs is $\tau_n$ and $\tau_n \ll T$. Before an MTCD attempts to access the v-gNB at each RAO, it will undergo an ACB mechanism with the barring probability broadcast by the corresponding v-gNB. At the $k$-th time slot, the MANO layer should allocate the limited PDCCH resources to each slice for the transmission of Message 2 and Message 4. We use $R^{Msg2}$ and $R^{Msg4}$ to denote the number of CCEs required to transmit one Message 2 and one Message 4, respectively.

We use the integer variable $r_n^k > 0$ to denote the number of CCEs lent by slice $n$ at the $k$-th time slot. In order to obtain resources from the Infrastructure Provider (InP), each slice needs to pay a certain price to the InP. For service differentiation, the resource price paid by each slice may be different. We use the price vector $\boldsymbol{\xi}^k = (\xi_1^k, \cdots, \xi_n^k)$ to denote the price that each slice pays to the InP at the $k$-th time slot. It should be highlighted that due to slice isolation, each MTCD can only request CCE resources from its associated slices. Therefore, we have the following capacity constraint:

$$\sum_{n \in \mathcal{N}} r_n \le R \tag{1}$$

As a consequence, a maximum of $r_n^k / (R^{Msg2} + R^{Msg4})$ Message 2's and Message 4's can be sent in the $k$-th slot. Our aim is to determine the optimal allocation vector $\mathbf{r}^* = (r_1^*, \cdots, r_n^*)$ to minimize the average delay of the MTCDs in each slice.

### 4.2. Traffic Model

In mMTC, the collision of the preamble is inevitable since the number of MTCDs is usually much larger than that of available preambles [29]. In each time slot, the number of backlogged requests $g_n^k$ includes both the newly arrived requests and the requests that failed in previous time slots. We assume that each MTCD remains backlogged until it successfully accesses the RAN. Similar to the analytical model in [30], we assume that

each v-gNB knows the value of $g_n^k$. Thus, the average number of MTCDs that succeed in transmitting sPreambles in slice $n$ can be expressed by [30]:

$$M^s(g_n^k) = \begin{cases} g_n^k \left(1 - \frac{1}{p_n}\right)^{g_n^k - 1}, g_n^k < p_n \\ p_n \left(1 - \frac{1}{p_n}\right)^{p_n - 1} \approx \frac{p_n}{e}, g_n^k \geq p_n \end{cases} \tag{2}$$

where $e$ is the Euler's number.

It should be highlighted that the approximation $(1 - 1/p_n)^{p_n - 1}$ is very accurate for $p_n = 54$, which is the typical number of preambles in a cell. In addition to preamble collision, the shortage of PPDCCH resources can also lead to RA failure. Since $r_n^k$ CCEs are allocated to slice $n$, the average number of MTCDs that succeed in accessing the RAN is given by [30]:

$$m^s(r_n^k) = \min\left\{\frac{p_n}{e}, \frac{r_n^k}{R^{Msg2} + R^{Msg4}}\right\}. \tag{3}$$

Since the time interval between two consecutive RAOs is $\tau_n$, the maximum rate of successful access can be expressed by

$$\vartheta_n^k = \frac{m^s(r_n^k)}{\tau_n}, \tag{4}$$

where $\vartheta_n^k$ is referred to as *channel capacity* in the literature [30].

In slice $n$, it is assumed that the RA requests conform to a uniform distribution over the $k$-th time slot. Especially, the rate of arrival is given by [30]:

$$\lambda_n^k(t) = \frac{M_n^k}{T}, (k-1)T \leq t \leq kT. \tag{5}$$

In mMTC, the access delay is one of the most important performance indicators of the RA procedure, which is defined as the time elapsed between the first RAO after its arrival and the last RAO at which the MTCD succeeds in accessing the v-gNB. According to whether the backlog is stable, the evaluation of access delay can be examined in the following two cases.

### 4.2.1. Stable Backlog

For a stable backlog, the average number of arrival access requests between two RAOs equals the channel capacity. In this case, the average number of successful MTCDs can be approximated by [30]:

$$g(x) \approx -p_n W(-\frac{x}{p_n}), \tag{6}$$

where $W(\cdot)$ is the principal branch of the Lambert W function, which is the inverse function associated with the equation $We^W = f(W)$ [30]; $x$ is the average number of arriving MTCDs between two RAOs; $g(x)$ is the average number of backlogged MTCDs. For the arrival rate given by (5), the average number of arriving MTCDs between two RAOs is $x = M_n^k \tau_n / T$. Thus, the expected delay is given by [30]:

$$d_n = \left(\frac{g\left(M_n^k \tau_n / T\right)}{M_n^k \tau_n / T} - 1\right)\tau_n, \text{ if } M_n \leq T\vartheta_n^k \tag{7}$$

### 4.2.2. Unstable Backlog

If the backlog is unstable, the number of arrival requests is higher than the channel capacity. In this case, the backlog increases with the arrival of the RA requests. The

backlog can be eliminated at time $t_{end} = M_n^k / \vartheta_n^k$. Thus, the expected success time is $E[t_{succ}] = M_n^k / 2\vartheta_n^k$.

The average arrival time of each MTCD in slice $n$ is $E[t_{arr}] = T/2$. Thus, the average delay of the MTCDs in slice $n$ is given by [30]:

$$d_n^k = E[t_{succ}] - E[t_{arr}] = \frac{M_n^k}{2\vartheta_n} - \frac{T}{2}, \text{ if } M_n^k > T\vartheta_n^k. \tag{8}$$

To summarize, the expected delay of the MTCDs in slice $n$ at the $k$-th time slot is given by:

$$d_n^k = \begin{cases} \left( \frac{g\left(M_n^k \tau_n / T\right)}{M_n^k \tau_n / T} - 1 \right) \tau_n, \text{ if } M_n \leq T\vartheta_n^k \\ \frac{M_n^k}{2\vartheta_n} - \frac{T}{2}, \text{ otherwise} \end{cases} \tag{9}$$

Thus, the expression of the expected delay is dependent on $\vartheta_n^k$, which in turn depends on the decision variable $\mathbf{r}_n^k$.

### 4.3. Problem Formulation and Analysis

We define the *weighted average access delay* (WAD) as:

$$u_n^k = \rho_n^k \cdot d_n^k, \tag{10}$$

where $\rho_n^k$ is the *priority factor* that indicates the importance of the access delay of slice $n$. A larger $\rho_n^k$ indicates a tighter latency requirement. In practice, the priority factor $\rho_n^k$ can be set according to the price vector $\boldsymbol{\xi}^k$. The effect of $\rho_n^k$ will be further examined in Section VI.

To summarize, DPRAP can be stated as follows: given the available PDCCH resources $R$, decide the optimal resource allocation variable $\mathbf{r}^1, \cdots, \mathbf{r}^k, \cdots$ such that the long-term average WAD is minimized, which can be formulated as

$$\min_{\mathbf{r}^0, \cdots, \mathbf{r}^K} \lim_{K \to \infty} \frac{1}{K} \sum_{k=1}^{K} \sum_{n=1}^{N} u_n^k, \tag{11}$$

$$s.t. \ (1) - (9), \tag{12}$$

$$r_n^k \in \mathbb{N}_+, \forall n \in \mathcal{N}, \forall k. \tag{13}$$

DPRAP is hard to solve due to the following difficulties. First, the expression of its objective function is uncertain. Whether the objective function is given by (7) or (8) depends on the optimization variable $\mathbf{r}^k$ itself. Second, it is a long-term objective that is unable to be resolved by traditional optimization theory. Moreover, DPRAP is a Mixed Integer Programming (MIP), which is an NP-hard problem. It is impossible to devise a polynomial time algorithm to solve it unless P=NP. Fortunately, the advent of RL has made it possible to resolve this problem. By interacting with the environment, the RL agent does not require any prior information about the specific objective function. Instead, it can learn to optimize the long-term objective according to a scalar (i.e., the reward) obtained by interactions with the environment. Thus, in this paper, we propose a learning-based algorithm to solve DPRAP, which is elaborated on in the subsequent sections.

## 5. Actor–Critic-Based Dynamic Resource Allocation Scheme

In this section, we first reformulate DPRAP as a Markov Decision Process (MDP). Then we design the ACRS by adopting the Actor-Critic framework to solve the MDP. Finally, we provide rigorous theoretical proof of the convergence of the proposed scheme.

### 5.1. The MDP Reformulation of DPRAP

DPRAP is essentially a sequential decision-making problem that requires the MANO entity to decide the optimal resource allocation at the beginning of each time slot, with the

aim of optimizing the long-term WAD. Such problems are hard to resolve with traditional optimization methods. Fortunately, MDP [31] provides a framework for modeling a sequential decision problem in dynamic environments. In MDP, the decision maker is called the *agent*, which learns the optimal policy through continuous interactions with the environment by trial-and-error. In practice, the learning of an MDP is both data-intensive and computation-intensive. Therefore, in our framework, the agent corresponds to the MANO entity that is deployed in conjunction with the gNB, as the gNB is capable of collecting essential samples as well as providing sufficient computing resources for the learning process.

An MDP is defined by a quintuple $\{S, A, P, R, \gamma\}$, in which $S$ is the state space; $A$ is the action space; $P$ defines the state transition probabilities; $R$ is the reward function; $\gamma$ is the discount factor. At the beginning of each time slot, the agent selects an action $a \in A$ based on its observation of the current state $s$ of the environment. As a consequence of this action, the agent receives a reward $r(s, a)$ and finds itself transiting to a new state $s' \in S$ with probability $P(s'|s, a)$. At the new state $s'$, the above interactions will repeat until the optimal policy is learned by the agent.

In the following, we will elaborate on the essential elements that define the MDP of DPRAP.

### 5.1.1. State Space

We define the state of the MDP by an $N$-dimensional vector, i.e.,

$$s(k) = (M_1^k, \cdots, M_N^k). \tag{14}$$

### 5.1.2. Action Space

Since the agent aims to decide the optimal PDCCH resource allocation, the action of the MDP is defined as:

$$a(k) = (r_1^k, \cdots, r_N^k). \tag{15}$$

To avoid resource monopoly and to further "compress" the action space, we require that the number of CCEs allocated to each slice is constrained by:

$$0 \le r_n^k \le \left\lfloor \frac{\xi_n^k}{\|\boldsymbol{\xi}^k\|} R \right\rfloor, \forall n \in \mathcal{N}, \forall k, \tag{16}$$

where $\|\boldsymbol{\xi}^k\|$ is the $L_1$ norm of vector $\boldsymbol{\xi}^k$.

### 5.1.3. State Transitions

The state transition $P$ defines the dynamics of the MDP, which specifies the *state transition probability* $p(s'|s, a)$ for each action $a$ under state $s$. Since both its action space and state space are finite spaces, the MDP is a finite MDP. Therefore, the state transition of the MDP can be represented by the *state transition probability matrix* $\mathbf{P}$. However, due to the stochastic arrival and departure of MTCDs within each slice, the explicit expression of $\mathbf{P}$ cannot be expressed directly. Therefore, the considered MDP is model-free.

### 5.1.4. Reward Function

In MDP, the reward is a scalar to be *maximized* by the agent. Since our goal is to *minimize* the WAD of MTCDs, we define the immediate reward of selecting action $a$ under the state $s$ as the negative of the average WAD of the MTCDs in all network slices:

$$r(s, a) = -\frac{1}{N} \sum_{n=1}^{N} \delta_n \cdot \tilde{d}_n^k, \tag{17}$$

where $\tilde{d}_n(k)$ represents the average access delay of the MTCDs within slice $n$ during the $k$-th time slot.

It should be emphasized that $\tilde{d}_n(k)$ is not evaluated according to the expression in (9). Instead, it is reported by the MTCDs that succeed in accessing the v-gNB in slot $k$. Suppose that at the $k$-th slot, the set of successful MTCDs in slice $n$ is $\tilde{\mathcal{M}}_n^k \subset \mathcal{M}_n^k$. Then, for each MTCD $n_i^k \in \tilde{\mathcal{M}}_n^k$, it can send its true access delay $\hat{d}_i^k$ to the v-gNB through Message 4. Thereafter, the agent can evaluate the average access delay of slice $n$ according to:

$$\tilde{d}_n^k = \frac{1}{\tilde{M}_n^k} \sum_{n_i^k \in \tilde{\mathcal{M}}_n^k} \hat{d}_i^k, \tag{18}$$

where $\tilde{M}_n^k$ is the cardinality of the set $\tilde{\mathcal{M}}_n^k$.

### 5.2. The AC-Based Resource Sharing Algorithm

There have been some well-known algorithms to solve an MDP such as dynamic programming. However, these algorithms are unable to solve model-free MDPs. Therefore, in this paper, we employ reinforcement learning approaches to solve the MDP. By combining the advantages of value-based and policy-based algorithms, the Actor-Critic (AC) algorithm has good convergence and small value-function estimation variance. Thus, we propose an AC-based dynamic resource sharing scheme, called ACRS, to solve this MDP. The pseudo-code of the proposed ACRS is shown in Algorithm 1. In Algorithm 1, the *decision epoch* is defined as the beginning of each time slot. As shown in Figure 3, the training process is mainly composed of four steps: action selection, mMTC random access, state value updating, and policy updating.

---

**Algorithm 1:** Actor-Critic-based Resource Sharing Scheme (ACRS) for DPRAP

---

**Input** : Parameterized policy $\pi_\theta(a, s)$; parameterized state-value function $V_w(s)$; step size $\alpha^\theta > 0$ and $\alpha^w > 0$; iteration limit $K_{max}$

1 Initialize policy parameter' and the state-value weights $w$ both to **0**;
2 Initialize $k = 0$;
3 Initialize $s$ to the first state of the MDP ;
4 **while** $k \leq K_{max}$ **do**
5     Sample $a \sim \pi_\theta(a, s)$ ;
6     Allocate the PDCCH resource to each slice according to action $a$ ;
7     Evalue the average access delay of the successful MTCDs in each slice according to (18);
8     Calculate the reward $r(s, a)$ according to (17) ;
9     The environment transits to the next state $s'$;
10     $\delta \leftarrow R + \gamma V(s') - V(s)$;
11     Update the state-value according to (21);
12     Update the parameterized policy according to (22);
13     $s \leftarrow s'$;
14     $k \leftarrow k + 1$
15 **end**

**Output:** The converged policy $\pi_\theta(a, s)$

---

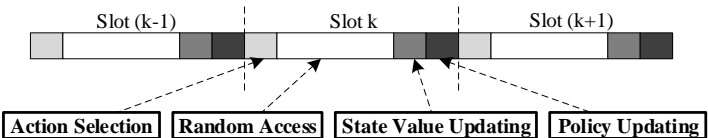

**Figure 3.** The procedure of ACRS.

### 5.2.1. Action Selection

At the beginning of the $k$-th time slot, let the state of the MDP be $s^k$. The agent chooses an action $a \in A$ with the probability of Boltzmann distribution, which is given by:

$$\pi_\theta = \frac{\exp(Q(s,a)/\theta)}{\sum_{b=1}^N \exp(Q(s,b)/\theta)} \tag{19}$$

where $\theta$ is the temperature parameter. $Q(s,a)$ is the obtained average reward for taking action $a^k \in A$ at state $s \in S$. Thus, the smaller value of $\theta$, the higher the probability of the action with a high average reward being selected, and vice versa. By taking the selected action $a^k$, each slice will be assigned a certain amount of PDCCH resources.

### 5.2.2. mMTC Random Access

After each slice obtains the allocated PDCCH resource, its subscribed MTCDs can request access to the corresponding v-gNB. The MTCDs that succeed in the RA procedure will report their access delays to the v-gNB, while the failed MTCDs' requests will be postponed to the subsequent time slots. According to (17) and (18), the agent can evaluate the reward of taking action $a^k$ in state $s^k$. Meanwhile, the MDP will transit to the next state $s^{k+1}$.

### 5.2.3. State-Value Function Updating

Based on the received reward $r^k(s^k, a^k)$, the Time Difference (TD) error can be calculated by:

$$\delta^k(s^k, a^k) = r^k(s^k, a^k) + \gamma V^k(s^{k+1}) - V^k(s^k). \tag{20}$$

The TD error will be fed back to the critic and the actor to update the state value function and parameterized policy, respectively. The state value function is updated according to

$$V^{k+1}(s^k) = V^k(s^k) + \mu(\tau_1(s^k, k)) \cdot \delta^k(s^k, a^k), \tag{21}$$

where $\tau_1\left(s^k, k\right)$ represents the number of times that state $s^k$ occurs in the $k$th access stage. $\mu(\tau_1(s^k, k))$ is a positive step parameter, which can affect the convergence rate of the algorithm. The value of $\mu(\cdot)$ will decrease with the occurrences of $s^k$. As the occurrence of the same state $s^k$ increases, the learning rate will decrease, and therefore, the convergence rate of the algorithm will accordingly decrease.

### 5.2.4. Policy Updating

After the state value is updated, the TD error will be used to update the actor's policy according to

$$\theta^{k+1}(s,a) = \theta^k(s^k, a^k) + \beta(\tau_2(s^k, a^k, k))\delta(s^k, a^k)\nabla log\pi_\theta(s^k, a^k), \tag{22}$$

where $\tau_2(s^k, a^k, k)$ is the number of executions of action $a^k$ at state $s^k$ during the $k$-th time slot. $\beta(\cdot)$ is a positive step parameter, which has the same meaning and effect as $\mu(\cdot)$ in (21).

### 5.3. Convergence Analysis

Next, we will theoretically prove the convergence of ACRS. According to the theorem mentioned in [32], the convergence of an AC-based algorithm could be ensured if the following assumptions hold.

**Assumption 1.** *The learning rate for actor and critic ($\beta_k$ and $\mu_k$) are positive, non-increasing, and also satisfy $\beta_k / \mu_k \to 0$, which requires that the time scale of the state value parameters update should be faster than that of the actor. Meanwhile, the following conditions should be satisfied:*

$$\begin{cases} \delta_k > 0, \forall k, \\ \sum_k \delta_k = \infty, \\ \sum_k (\delta_k)^2 < \infty, \end{cases} \tag{23}$$

*where $\delta_k$ could be either $\beta_k$ or $\mu_k$.*

**Assumption 2.** *For $\forall \theta \in R^n$, the Markov chains $\{S_k\}$ and $\{S_k, A_k\}$, which correspond to the transition probability P, are irreducible and aperiodic under the Random Selection Policy (RSP) $\pi_\theta(s, a)$ given by* (19).

**Assumption 3.** *For $\forall s \in S, \forall a \in A$, the RSP $\pi_\theta(s, a)$ is twice differentiable to the parameter $\theta$. Meanwhile, the condition $\pi_\theta(s, a) > 0, \forall \theta \in R^n, \forall s \in S, \forall a \in A$ should be satisfied.*

**Assumption 4.** *Define the matrix $G(\theta)$ as*

$$G(\theta) = \sum_{s,a} \eta_\theta(s, a) \psi_\theta(s, a) \psi_\theta(s, a)^T, \tag{24}$$

*where*

$$\psi(s, a) = \frac{\nabla \pi_\theta(s, a)}{\pi_\theta(s, a)} = \nabla ln \pi_\theta(s, a), \tag{25}$$

*and*

$$\eta_\theta(s, a) = \pi_\theta(s) \cdot \pi_\theta(s, a), \tag{26}$$

*$G(\theta)$ should be uniformly positive definite, which means that there exists some $\epsilon_1 > 0$ such that*

$$r^T G(\theta) r \geq \epsilon_1 \|r\|^2, \forall r \in R^m, \forall \theta \in R^n. \tag{27}$$

**Theorem 1.** *In the proposed ACRS, we have*

$$\liminf_k \|\nabla \lambda(\theta_k)\| = 0, w.p.1. \tag{28}$$

*Furthermore, if $\theta_k$ is bounded w.p. 1, then*

$$\lim_k \|\nabla \lambda(\theta_k)\| = 0, w.p.1, \tag{29}$$

*where $\lambda(\theta_k) = \sum_{S,A} r(s, a) \pi_\theta(s) \pi_\theta(s, a)$.*

We will show that our proposed algorithm satisfies the above four assumptions. In our MDP model, the learning rate for critics and actors is defined as $\beta_k = \mu(\tau_1(s^k, k))$ and $\alpha_k = \mu\left(\tau_2\left(s^k, a^k, k\right)\right)$, respectively. The function $\mu(\cdot)$ is a non-negative monotonically increasing function. $\tau_2(s^k, k)$ is the number of occurrences of the state $s^k$. $\tau_2(s^k, a^k, k)$ is the number of executions of action $a^k$ in state $s^k$. Thus, $\tau_2\left(s^k, a^k, k\right) < \tau_2(s^k, k)$. Therefore, Assumption 1 is satisfied.

For Assumption 2, we notice that in the considered MDP, all states transit based on the Boltzmman RSP. Meanwhile, the whole learning process is episodic and the system will be reinitialized at the end of every episode. Thus, the MDP is irreducible and aperiodic [33].

Next, we verify Assumption 3 by differentiating the RSP $\pi_\theta(a)$ with respect to the parameter $\theta$. Let

$$H(s, a) = \exp(Q(s, a)/\theta), \tag{30}$$

and we can obtain

$$\nabla \pi_\theta(s,a) = H(s,a) \left[ \sum_{b=1}^{N} Q(s,b)H(s,b) - Q(s,a) \sum_{b=1}^{N} H(s,b) \right] / \left[ \theta \sum_{b=1}^{N} H(s,b) \right]^2 \quad (31)$$

It can be seen that $H(s,a)$ is a part of polynomials, which is exponential. Thus $\nabla \pi_\theta(s,a)$ is differentiable and the adopted Boltzmman RSP is twice differentiable. Therefore, Assumption 3 holds.

In our algorithm, we linearly approximate the state value function as

$$V_v^\theta(s,a) = \sum_{i=1}^{M} v^i \phi_\theta^i(s,a), \quad (32)$$

where $\mathbf{v} = (v^1, \cdots, v^M)$ and $\boldsymbol{\phi}_\theta = (\phi_\theta^1(s,a), \ldots, \phi_\theta^M(s,a))$ are the parameter vector and feature vector of the state value function, respectively. In ACRS, the feature vector could be obtained by

$$\phi_\theta^i(s,a) = \delta^i \psi_\theta(s,a), \quad (33)$$

where $\psi_\theta(s,a)$ can be computed by dividing the derivatives by $\pi_\theta(s,a)$. $\delta^i$ is a priority factor to balance the differences of all network slices. Thus, the features corresponding to these parameters will be linearly independent across the state space.

We rewrite function $G(\theta)$ in the matrix form as the following:

$$G(\theta) = \phi_\theta(s,a)^T \eta_\theta(s,a) \phi_\theta(s,a). \quad (34)$$

Thus,

$$r^T G(\theta) r = r^T \phi_\theta(s,a)^T \eta_\theta(s,a) \phi_\theta(s,a) r > 0. \quad (35)$$

As the columns of $\phi_\theta(s,a)$ are linearly independent, then $\phi_\theta(s,a)r \neq 0$. Therefore, we conclude that $G(\theta)$ is positive definite. We have

$$\inf_{r \neq 0} \frac{r^T G(\theta) r}{\|r\|^2} = \inf_{\|r\|=1} \frac{r^T G(\theta) r}{\|r\|^2} \cdot \inf_{\|r\|=1} r^T G(\theta) r. \quad (36)$$

Since the set $F = \{r : \|r\| = 1\}$ is compact and $r^T G(\theta) r$ is continuous in $r$, the minimum of the left-hand side expression in (36) can be achieved within the set $F$. Similar to the proof in [34], we can conclude that $G(\theta)$ is uniformly positive definite for all $\theta$. Thus, Assumption 4 holds.

To summarize, Proposition 1 could be drawn and can be used to prove the convergence of ACRS.

**Proposition 1.** *According to Theorem 2, based on any given policy $\pi'$, the average reward and the parameter vector of the state value function converges to the q-function $Q_{\pi'}(s,a)$ and $v(\pi')$, respectively, with probability 1, provided that Assumptions 1, 2, 3, and 4 are satisfied.*

## 6. Numerical Results

In this section, we evaluate the performance of our proposed network slicing-enabled RA framework through simulation experiments. First, we examine the convergence of ACRS. Then we show the performance of ACRS under different network parameters, including the amount of available resources, the number of MTCDs, and the priority factor. Finally, we evaluate the performance gain of the proposed framework by comparing it with benchmark algorithms.

### 6.1. Simulation Settings

The considered scenario is shown in Figure 1. For ease of reference, we summarize the key parameters in Table 2. The considered physical network is a single-cell mobile

network with a radius of 1 kilometer, which is composed of one gNB and a number of randomly distributed MTCDs. On the gNB side, the number of available preambles for contention-based RA is 54. Meanwhile, the number of available PDCCH resources of the gNB is up to 25 CCEs. The RA requests of MTCDs in each slice conform to uniform distribution, as shown in (5). In the simulation, let there be five different types of mMTC applications, each of which is supported by a dedicated network slice. According to [35], the initial proportion of MTCDs with different QoS requirements is set to be 40%, 20%, 20%, 10%, 10%, respectively.

Furthermore, we set the priority factor ratio of these five network slices to 5:4:3:2:1. According to [36], the packet size of the RA request initiated by MTCD is set to 1KB-1MB. The transmit power of MTCD is 100 mW. The power of the background noise is assumed to be 1 mW, and the path loss of mMTC communication is $8 + 37.6 \log_{10}(d(m))$ [28]. For simplicity, the number of sPreambles in each slice is identical, which is set to 54.

**Table 2.** Parameters of the simulations.

| Parameter | Value |
| --- | --- |
| Simulation duration | 50–100 time slots |
| Radius of the network | 1km |
| Number of MTCDs | 50 to 5000 |
| RA requests distribution | Uniform distribution |
| Number of network slices $N$ | 5 |
| Proportion of mMTC services | $4 : 2 : 2 : 1 : 1$ |
| Priority factor ratio of the network slices | $5 : 4 : 3 : 2 : 1$ |
| Packet size of the access requests | 1KB/500KB/1MB |
| Transmit power of MTCDs | 100 mW |
| Power of background noise | 1 mW |
| Path loss | $8 + 37.6 log_{10}(d(m))$ |
| Number of CCEs of PDCCH resources | 25 |
| Number of preambles | 54 |
| Number of sPreambles | $54 \times N$ |

*6.2. Simulation Results and Discussions*

6.2.1. Convergence Performance

In Figure 4, we show the convergence of the proposed ACRS under different resource conditions. In this simulation, we set the number of MTCDs in the network to be $M = 500$. The number of PDCCH resources $R$ is set to be 10, 15, 20, and 25 CCEs, respectively. We run the simulation for 70 time slots. Since the reward of the MDP is the negative of the average WAD, we multiply it by $-1$ to convert it to the average WAD for the sake of clarity.

It can be seen that the average WAS represented by the four curves tends to stabilize after a certain period of training. This result demonstrates that our proposed ACRS has a fast convergence speed. In addition, we can observe that the training time of ACRS increases with the number of available resources. In particular, it only needs about 9 time slots to converge when $R = 10$, while 36 time slots are required in the case of $R = 25$. This is because as the number of PDCCH resources increases, the action space of the MDP becomes larger. Therefore, more training time slots are required to explore a larger action space.

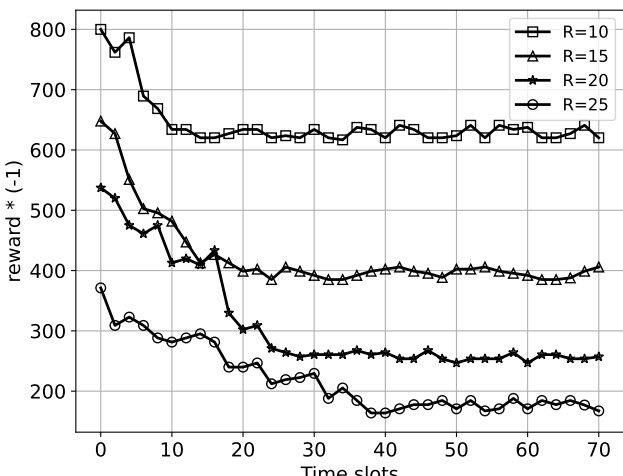

**Figure 4.** Convergence of the proposed ACRS under different resource conditions.

6.2.2. Performance under Different Parameters

In Figure 5, we plot the average access delay as a function of the number of available PDCCH resources. In this experiment, we set the number of MTCDs as $M = 5000$. The number of PDCCH resources varies from 5 to 25 CCEs. The results are averaged over 50 time slots. As expected, it is observed that the average access delay decreases with the amount of available PDCCH resources. Furthermore, we can also observe that the average access delay of Slice 1 is much lower than that of Slice 5. Thus, this result demonstrates that the proposed framework can guarantee the QoS of delay-sensitive MTCDs regardless of the available PDCCH resources. Meanwhile, this framework can provide the best-effort service for delay-tolerant MTCDs. This is the main advantage of introducing network slicing for supporting mMTC.

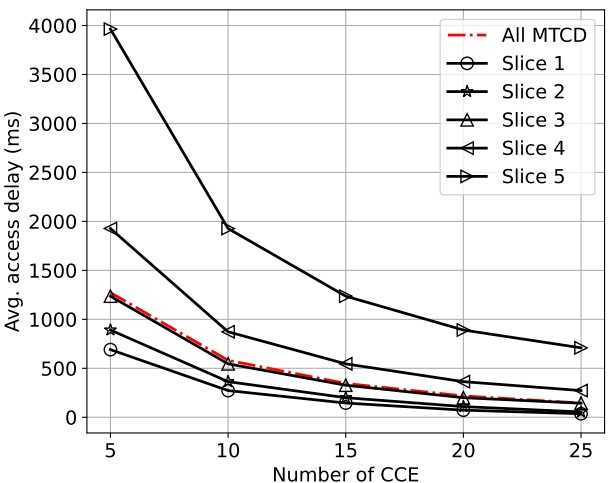

**Figure 5.** Average access delay vs. the number of available CCEs in PDCCH.

In Figure 6, we plot the average access delay of the proposed framework under different numbers of MTCDs. In this simulation, the available PDCCH resources are set to $R = 25$ CCEs. The number of MTCDs in the system varies from 1000 to 5000. Similar to the previous experiment, the simulation results are averaged over 50 time slots to eliminate randomness. We can see that the average access delay increases with the number of MTCDs. This is because when the amount of available resources is fixed, the probability of preamble collision and RA failure increases with the number of MTCDs, resulting in more frequent backoffs. In addition, we can also observe that the curve of delay-sensitive MTCDs is the lowest. This result verifies that our proposed framework can provide differentiated services for MTCDs with heterogeneous delay requirements.

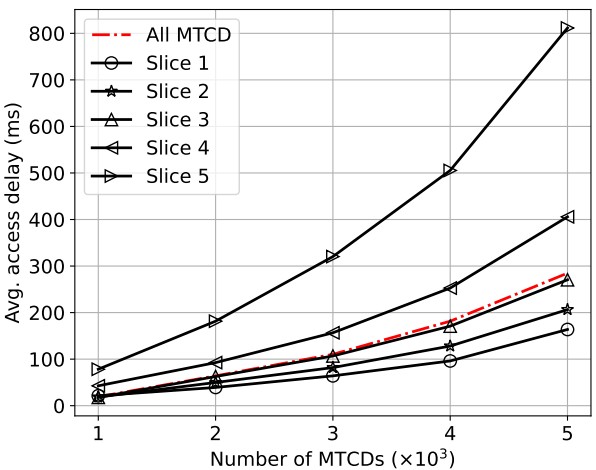

**Figure 6.** Average access delay vs. the number of MTCDs.

In Figure 7, we examine the effect of the priority factor on average access delay. In this experiment, we set the number of MTCDs as $M = 5000$, and the available PDCCH resources as $R = 25$ CCEs. The priority factors of the network slices are set according to the following three mechanisms.

- Ordinal priority: First, we sort the network slices in descending order according to the resource price (i.e., $\xi_n^k$) they paid to the InP. Then we successively assign the priorities $1, 2, \cdots, N$ to individual slices.
- Exponential priority: Similar to the ordinal priority mechanism, we first sort the network slices in descending order according to $\xi_n^k$. Then we successively assign the priorities $1, 2, 4, \cdots, 2^N$ to individual slices.
- Proportional priority: The priority of slice $n$ is proportional to the resource price it paid to the InP, i.e., $\delta_n^k = \xi_n^k / (\sum_{l \in \mathcal{N}} \xi_l^k)$.

Obviously, the proportional mechanism may lead to the highest variance of priority. In this experiment, we run the simulation for 50 time slots, and the results are averaged. From Figure 7, we can observe that the average access delay of ordinal priority and proportional priority mechanisms are almost the same. In contrast, by exploiting the exponential priority mechanism, the access delay of delay-tolerant MTCDs is very large, which suggests that most of them cannot access the RAN successfully. Thus, the priorities with moderate variance, such as ordinal and proportional mechanisms, are preferable.

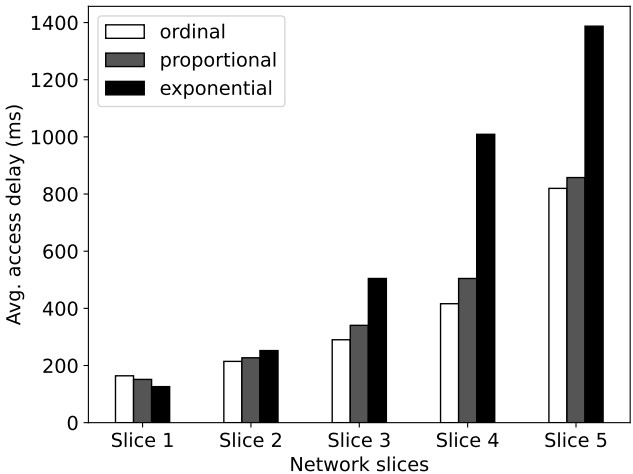

**Figure 7.** The effect of priority factors on the average access delay.

6.2.3. Comparison with Benchmarks

In this experiment, we compare the performance of our proposed ACRS with the following two benchmark algorithms:

- **Proportional Resource Allocation (PRA):** In this algorithm, the amount of resources allocated to each slice is proportional to the number of MTCDs associated with it. Formally, $\mathbf{r}^k$ is given by:

$$r_n^k = \frac{M_n^k}{\sum_{n=1}^N M_n^k} R. \tag{37}$$

- **Greedy:** In this algorithm, each slice *greedily* requests PDCCH resources to minimize their instantaneous average delay at each time slot.

In Figure 8, we plot the average access delay of ACRS and the benchmark algorithm for different numbers of available PDCCH resources. In this experiment, we set the number of MTCDs as $M = 2500$, and the available PDCCH resources are set as $R = 5, 15, 25$ CCEs. We run the simulation for 50 time slots. The average access delay of each network slice is evaluated. The simulation results show that when the available resources are insufficient (e.g., when $R = 5$), ACRS can preferentially guarantee the QoS of delay-sensitive slices. In contrast, the benchmark algorithms cannot provide QoS guarantees for delay-sensitive slices, and the delay is much larger than that of ACRS. In the case of sufficient PDCCH resources (e.g., when $R = 25$), the average access delay of these algorithms is similar. This is because in the case of sufficient resources, these algorithms can allocate enough resources to each slice. Therefore, the access delay is mainly decided by the backoff delay that depends on the number of available sPreambles in each slice. Since each slice has the same number of sPreambles, their average access delay is almost equal.

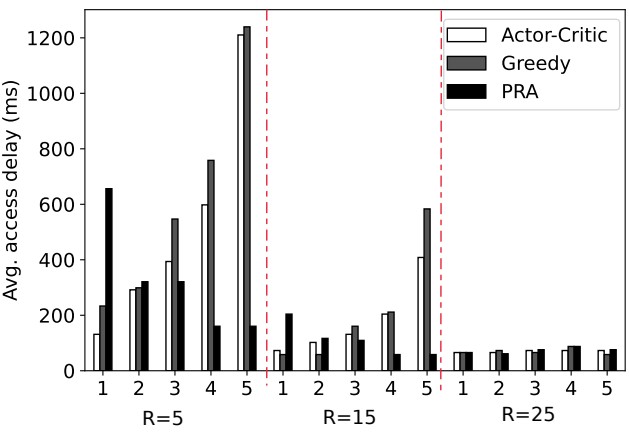

**Figure 8.** Performance comparison with benchmark algorithms in terms of average access delay. Slice 1 is delay-sensitive, while Slice 5 is delay-tolerant.

## 7. Conclusions and Future Work

In this work, we have proposed a network slicing-enabled RA framework for mMTC networks. In order to alleviate the resource scarcity during the RA procedure, we have proposed the concept of sPreamble and a learning-based resource sharing scheme. Simulation results show that the proposed framework can effectively allocate resources for each network slice to ensure the QoS requirements of the MTCDs in RA. The proposed framework is effective in meeting the rapidly growing RA demands for mMTC applications in ultra-dense networks. In addition to improving the performance of the RA procedure, our proposed framework can also guarantee the differentiated QoS requirements for various MTCDs.

In this paper, a network slicing-based RA framework is proposed, and its performance is preliminarily verified by numerical simulation. However, this paper does not provide

theoretical analyses of the performance gains of the proposed framework, including its performance in mitigating collision probability and RAN congestion, etc. These could be interesting issues to be investigated in our future work.

**Author Contributions:** Conceptualization, Z.J.; Methodology, J.W.; Resources, P.C.; Data curation, X.S.; Writing—original draft, B.Y. and F.W.; Writing—review and editing, B.Y. and F.W.; Supervision, B.Y., X.S. and J.Z.; Project administration, B.Y.; Funding acquisition, B.Y. All authors have read and agreed to the published version of the manuscript.

**Funding:** This research received no external funding.

**Data Availability Statement:** Not applicable.

**Acknowledgments:** This work was supported by China Telecom Research Institute.

**Conflicts of Interest:** The authors declare no conflict of interest.

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
