# Peer review of "Intelligent Random Access for Massive-Machine Type Communications in Sliced Mobile Networks"

_electronics, doi:10.3390/electronics12020329_

Round 1
Reviewer 1 Report
The authors in this paper discus the RA in 5G and the problem is modeled with MDP and its formulated explicitly. So, some remarks needs to be clarified :
1. What is the Law for the arrival and departure paquets
2. The processus say in this paper is irreducible and aperiodic, where is the proof
3. Equation (21), change state s by s^k
Reviewer 2 Report
Comments to the Author:
The authors propose a network slicing-enabled intelligent random access framework for massive Machine-Type Communication (mMTC). A massive physical network sliced into several lightweight networks to reduce collision and improve QoS. A novel concept of sliced preambles is also proposed.
Though the paper is readable, and the math is sensibly presented, the paper has some issues that I feel need to be addressed carefully, as follows:
1. In abstract, line no. 12, ‘whose number scales linearly with the number of network
slices’ --- is not sounding well.
2. Equations (3), (5), (7), (8) etc. are your contributions? If not please mention the references.
3. The convergence of the proposed scheme is verified by Machine Learning algorithm. The source of data set is not mentioned.
4. There is no comparative result. How you can claim that the proposed technique outperforms the existing techniques. You can take any one reference paper from your manuscript (better from [7-13] ) and compare you obtained result.
5. The value of noise you considered is 1 mW, i.e. 0 dBm, it is high. Normally, the noise power is considered around 70 dBm (approx..).
6. Some typo errors are there, so I suggest the text to be revised.
Reviewer 3 Report
I would like to endorse the publication of this manuscript.
